# Measuring electrical properties of the lower troposphere using enhanced meteorological radiosondes

R. Giles Harrison

Department of Meteorology, University of Reading, Reading, RG6 6BB, UK

**Abstract**. In atmospheric science, measurements above the surface have long been obtained by carrying instrument packages, radiosondes, aloft using balloons. Whilst occasionally used for research, most radiosondes - around one thousand are released daily - only generate data for routine weather forecasting. If meteorological radiosondes are modified to carry additional sensors, of either mass-produced commercial heritage or designed for a specific scientific application, a wide range of new measurements becomes possible. A programme to develop add-on devices for standard radiosondes, which retains the core meteorological use, is described here. Combining diverse sensors on a single radiosonde helps interpretation of findings, and yields economy of equipment, consumables and effort. A self-configuring system has been developed to allow different sensors to be easily combined, enhancing existing weather balloons and providing an emergency monitoring capability for airborne hazards. This research programme was originally pursued to investigate electrical properties of extensive layer clouds, and has expanded to include a wide range of balloon-carried sensors for solar radiation, cloud, turbulence, volcanic ash, radioactivity and space weather. For the cloud charge application, multiple soundings in both hemispheres have established that charging at the boundaries of extensive layer clouds is widespread, and likely to be a global phenomenon. This paper summarises the Christiaan Huygens medal lecture given at the 2021 European Geosciences Union meeting.

Keywords: electrostatics; dust; cloud; space weather; natural hazards; turbulence;

## 1.    Introduction and scientific motivation

This paper is based on material presented in my Christiaan Huygens medal lecture at the 2021 meeting of the European Geosciences Union. The original lecture was called "Perspicacity…and a degree of good fortune: opportunities for exploring the natural word". This title was inspired by Christiaan Huygens' own words reflecting on scientific progress in 1687:

"…difficulties…cannot be overcome except by starting from experiments... much hard work remains to be done and one needs not only great perspicacity but often a degree of good fortune". (Huygens, 1687)

Huygen's contention that both luck and insight are a critical combination in scientific progress was far-sighted. It feels especially relevant to experimental atmospheric science, in which the circumstances are entirely beyond the control of the experimenter. This paper describes some attempts to confront this and other challenges in exploring electrical properties of the lower atmosphere, with a particular focus on measuring the electric charge associated with extensive layer clouds. Unlike thunderclouds, which can separate substantial charges, the charge on layer clouds (e.g. stratus or stratocumulus clouds) is very weak and hence the signals to be investigated are small. Layer clouds are, however, relatively abundant, and play a role in the electrical balance within the Earth's atmosphere (Harrison et al, 2020) as well as in the energy balance of the climate system. To investigate them, sensors, instruments, platforms and the interpretation of indirect or related measurements are all required.

Progress in making related instruments and measurements is described here, with co-workers at the University of Reading. This programme has applied modern electronic methods to one of the oldest experimental topics in atmospheric science. New measurements aloft have been obtained by enhancing standard meteorological balloon systems and, more recently, by instrumenting uncrewed aircraft. This paper describes the principles of the measurement technology (section 2), the application to extensive layer clouds (section 3), and reviews the applications beyond atmospheric electricity, to radioactivity, space weather, turbulence, dust electrification and optical cloud detection (section 4). The overall findings concerning layer clouds are summarised in section 5, with general conclusions on the value of the enhanced radiosonde strategy given in section 6. Initially, to provide context and motivation with which to close this introductory section, early historical developments in atmospheric electricity and electrostatics are briefly described, followed by outlining the scientific questions around the possible relationship between space weather, ionisation and clouds.

### 1.1 Early atmospheric electrostatics

A convenient starting point is the defining year in atmospheric electricity, 1752, which is associated with Benjamin Franklin's famous kite experiment. Some exact details remain debated, but it provoked wider investigations of cloud electricity (Berger and Ait Amar, 2009). Less well known, however, are the findings about the electricity of fair

weather and non-thunderstorm clouds, which emerged around the same time. For example, by 1753, the pioneer investigator John Canton[1] had observed that

"The air without-doors I have sometimes known to be electrical in clear weather." (Canton, 1753).

For this, Canton had devised his own detection device for experiments, an electroscope (Figure 1), which operated by repulsion or attraction between charges induced on lightweight pieces of orange pith. With this apparatus, Canton determined the charge and polarity of clouds overhead, by comparing reference electrical effects generated from amber and sealing wax.

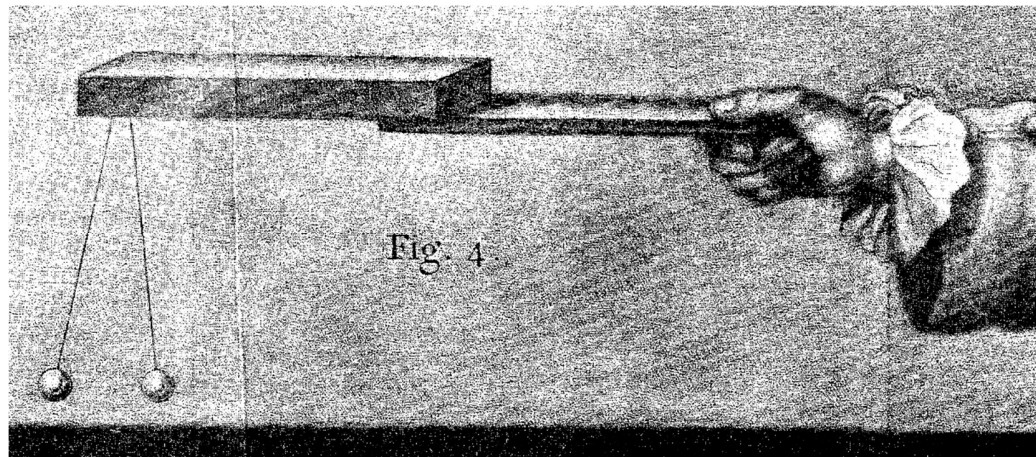

**Figure 1. The pith ball electrometer described by John Canton in 1753, to detect electrical changes (from Canton, 1753).**

Since Canton's time, the major topics of research in atmospheric electricity have been the study of lightning as a natural hazard and the quest to understand the fundamental origin of the electric field observed in fair weather[2], now understood to be due to continuous distribution of charge by the global atmospheric electric circuit. In this, thunderclouds have received the main attention. Other cloud types have nevertheless been suggested to be electrically influenced, for example in the treatise by Luke Howard (Howard, 1843), whose cloud classification system is still in widespread use. It is therefore not surprising that the pioneering nineteenth century balloonists James Glaisher and

---

1. John Canton (1718-1772) was a natural philosopher who experimented with electricity, magnetism and the properties of water. He received Copley medals of the Royal Society in 1751 and 1764. Canton was born in Stroud, Gloucestershire and worked and died in London. (Having been born and schooled in the same town, I am, perhaps, more aware of Canton's work than most).

2. "Fair weather" has a specific meaning in atmospheric electricity, in identifying a situation when no substantial convectively driven charge separation occurs locally (e.g. Harrison and Nicoll, 2018). In meteorology more generally, fair weather clouds are small and often numerous cumulus clouds, in an otherwise clear sky.

Henry Coxwell[3] carried an electrometer on their flights, with the electrical and meteorological measurements ultimately published together (Glaisher, 1862). Studying the physical behaviour of individual charged droplets also has a long history. For example, Lord Kelvin (William Thomson) calculated the electrical forces between droplets which are charged and free to polarise (Thomson, 1853), and Lord Rayleigh (John Strutt) observed that charged droplets were more inclined to coalesce than neutral droplets (Strutt, 1879).

**1.2 Weather and ionisation**

A key scientific motivation in clouds and atmospheric electricity is establishing whether the electrical interactions which constantly occur between ions, aerosols and drops can yield material effects in the atmosphere, and, especially, within clouds. This question presented itself during my PhD work on the charging of radioactive aerosols[4], provoked by the radioactive aerosols observed following the Chernobyl disaster and transported across Europe in weather systems. Theory showed that such aerosols would become charged through the emission of decay particles and the collection of ions (Clement and Harrison, 1992)[5]. Investigating other effects of radioactivity on weather, from releases of radioactive gas during nuclear reprocessing, raised further questions about electric charge effects (e.g. Harrison and ApSimon, 1994). Both topics illustrated the need for more experimental and theoretical research on electrical aspects of non-thunderstorm clouds.

Increased attention on the relationship between ionisation, aerosols, charge and droplet behaviour followed from the correlation published between global satellite retrievals of cloud and galactic cosmic ray (GCR) variations (Svensmark and Friis-Christensen, 1997). This opened a vigorous discussion on whether cosmic rays could directly influence clouds and climate. Although the detailed technical aspects fell between the conventional boundaries of atmospheric science, aerosol science and high energy physics, this did not prevent confident opinions being expressed. A possible series of processes linking GCR variability into weather phenomena through enhancement of droplet freezing by charged aerosols had in fact previously been suggested by Brian Tinsley (Tinsley and Deen, 1991), building on the considerable solar cycle modulation of lower atmosphere ionisation which had been recognised by Ney (1959) and Dickinson (1975). However, these papers - and indeed our own (Carslaw, Harrison and Kirkby, 2002, hereafter CHK02, and Harrison and Carslaw 2003) - also highlighted the limited knowledge of charge in non-thunderstorm atmospheric processes. CHK identified two physical routes linking high energy ionisation and clouds for further investigation, the "ion-aerosol clear-air" mechanism, leading to the formation of new cloud condensation nuclei (CCN), and the "ion-aerosol near-cloud" mechanism, leading to enhanced droplet charges. The strongest correlation with GCR was later identified to be with low level liquid water cloud (e.g. Marsh and Svensmark, 2000). Building on

3. These heroic measurements were brought to a wide audience through the 2019 film *The Aeronauts*.

4. This provided a fine introduction to atmospheric electricity, alongside the wonderful textbook of J.A. Chalmers (Aplin, 2016). It also indicated that the whole topic was overdue for new experiments.

5. The Clement-Harrison theory was confirmed by independent experiments (Gensdarmes et al, 2001). Wet removal of radioactive aerosols was found to be enhanced by their charge (Tripathi and Harrison 2002).

this, a proposal was made to the CERN laboratory to begin an entirely new seam of experimental work (Fastrup et al, 2000) - the "Cosmics Leaving OUtdoor Droplets" or CLOUD experiment. This international proposal was exciting to contribute to initially, although the final facility did not begin operation until 2009. CLOUD has since explored ion-induced aerosol nucleation in impressive detail, by firing a controlled energetic proton beam into an exquisitely instrumented experimental chamber.

An important outcome of the CLOUD experiment is the conclusion that variations in CCN from GCR, ie the CHK02 "ion-aerosol clear-air" mechanism, are too weak to influence clouds and climate (Pierce, 2017). In comparison, CHK02's "ion-aerosol near-cloud" mechanism has received far less attention, perhaps because the atmospheric situation is much less able to be represented by laboratory investigations. Such gaps in understanding of the detailed behaviour of clouds[6] are undesirable because the associated potential contributions to climate remain unquantified. This allows extravagant claims to be made where caution is more appropriate (e.g. Harrison et al, 2007). As some of the atmospheric electricity equipment originally developed for surface use seemed highly suitable for filling the gap in providing the relevant in situ measurements required, this encouraged me, quite possibly too enthusiastically, to propose undertaking my own "...experiments with weather balloons" (Pearce, 1998).

Much of the instrumentation, techniques, measurements and analysis described here follow from pursuing this apparently well-defined, but technically surprisingly difficult, scientific goal.

## 2.    Electrostatic measurements and instrumentation

Measurements of cloud and droplet charge require appropriate sensors combined with registering or recording devices. In general, whilst electroscopes simply indicate the presence of charge, electrometers are measuring instruments capable of registering exceptionally small charges and currents, or able to provide voltage measurements whilst drawing negligible current and therefore with minimal loading of the source. Measurements based on either mechanical or electronic principles are possible.

### 2.1 Mechanical

Mechanical detectors, such as the pith ball electroscope of Canton or indicating devices which used straw or gold leaf, combined the sensing and registering aspects, providing a visible response to the electric force. Probably the earliest identifiable example of an instrument employing this principle is the versorium of William Gilbert,

---

6. Extensive layer clouds would not be considered fair weather meteorologically, but, electrically, they would usually fulfil the conventional criteria. To try to avoid confusion between "fair weather" in the meteorological and atmospheric electrical usages, whilst retaining the important electrical distinction with disturbed weather, overcast extensive layer cloud circumstances are described here as semi-fair weather conditions.

"…make yourself a rotating needle electroscope, of any sort of metal, three or four fingers long, pretty light
and poised on a sharp point of a magnetic pointer." (Gilbert, 1600).

A later example of the electric force approach was the delicate torsion electrometer developed by Lord Kelvin, also
likely to be the device loaned to James Glaisher. To this electrometer, Kelvin added a sensor able to obtain the air's
local electric potential, through charge transfer of water drops falling from an insulated tank. By projecting the
electrometer's deflection onto photographic paper, the "water dropper" and electrometer combination made
continuous recording of the atmospheric electric field possible (Aplin and Harrison, 2014)[7].

Mechanical deflection technologies remain useful and were used in the twentieth century for atmospheric electricity
measurements (e.g. Wilson, 1908) and in the discovery of cosmic rays (Hess, 1912). Deflection also provided an
experimental method to determine droplet charge on the Millikan principle, by photographing the motion of falling
drops in an electric field (Tellus, 1956; Allee and Phillips, 1959).
**2.2 Electronic**
Mass-produced sensors are now typically integrated within chips providing amplification and a standard digital
interface, but for low volume science research sensors, implementing customised analogue signal conditioning
circuitry is still necessary. This is especially the case in electrometry, where the packaging of the parts is a critical
aspect because of the leakage currents which can arise. Early electronic methods depended on thermionic valves, in
general making the electrometer part of current flow in a circuit or across which a voltage is developed.

Electrometers are now readily constructed using modern semiconductors, in particular operational amplifiers (or op
amps) which, from their origins in performing mathematical operations, provide a very large multiplication of the
voltage difference between two input terminals. If the op amp is selected to have an especially small input bias current
(~1 fA) - and such devices are often specifically marketed as electrometer op amps - very small currents of only 1000s
of electronic charges per second or even less become measurable. With a particularly simple circuit configuration, an
op amp can be used to measure the voltage developed across an ultra-high source resistance, such as air in fair weather.
Typical electrometer op amps generally only have a relatively small dynamic range, but circuit configurations can
considerably extend this (e.g. Harrison 1996).

An op amp can be used to convert a current to a voltage, by adding a single feedback resistor in a "transresistance"
circuit (Figure 2a). The advantage of this configuration over just measuring the voltage across the resistor is that the

---

7. The international use and longevity of this technology is remarkable, providing measurements on the Eiffel Tower
during the 1890s (Harrison and Aplin, 2003), and at the UK's Kew and Eskdalemuir Observatories from 1861-1931
and 1908-1936 respectively. The water dropper atmospheric potential sensor at Kakioka Observatory in Japan only
ceased operation in 2021.

circuit's input is always at the same potential, essentially the circuit ground, which ensures the loading of the current
source remains constant whatever current is flowing. A practical difficulty with such circuits is in maintaining
scrupulous insulation, to prevent errors from leakage currents swamping the measurement current. It can also be
troublesome to obtain and calibrate suitably large value resistors[8], e.g. a $10^{12}$ Ω feedback resistor is needed to convert
$10^{-12}$A (i.e. 1 pA) to 1 Volt. Figure 2b shows the physical implementation of a current to voltage converter using
through-hole technology electronic parts. This device was powered by internal button cells, and constructed entirely
within a screened box. The input current connection was air-wired (i.e. positioned above the circuit board) to minimise
leakage. A second op amp allowed the gain to be adjusted to compensate for inaccuracies in the $10^{9}$ Ω feedback
resistor, using a ratiometric matching method implemented with readily obtained smaller value precision resistors
(Harrison, 1995a).

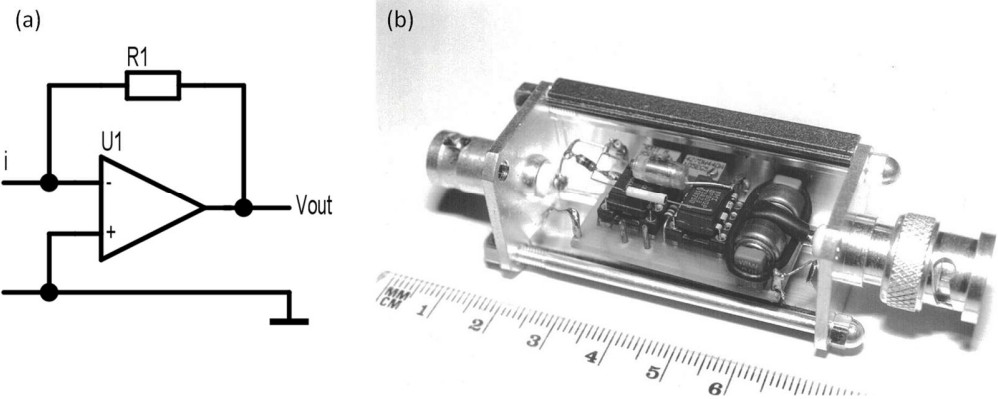


**Figure 2. Current measurements. (a) Principle of a current-to-voltage converter based on an operational amplifier (U1)**
**and a resistor (R1). (b) Implementation of a battery powered current-to-voltage converter, built within a small in-line case,**
**with the input current (~pA) presented on the left connector and the voltage output (~mV) on the right. (The tubular air-**
**wired component in front of the polystyrene capacitor is a high value resistor, From Harrison, 1995a.)**
Further refinements to these basic measurement themes have been needed, e.g. to extend electrometer voltmeter
measurements from about 10 V into the range 100 V to 10 kV (Harrison 1995b;1997;2000), to reject 50 Hz
interference (Harrison, 1997), to permit computer control of switching between current and voltage (Harrison and
Aplin, 2000a) and to implement a logarithmic response across a wide range of currents (Marlton et al, 2013). Methods
of calibration are also an important aspect, such as by bridge ratio methods for resistances (Harrison, 1997), or
differentiating a steadily changing voltage to generate a defined small current (Harrison and Aplin, 2000b).

---

8. An alternative is to synthesise the large resistance needed by combining a smaller feedback resistor with a resistive
divider, in a so-called "T network" (e.g. figure 3.14 in Harrison, 2014).

## 2.3 Examples of surface instruments

The importance of these electronic approaches is that they provide inexpensive routes to measure weak signal sources in environmental conditions without the need to put laboratory grade equipment at risk. They can be used directly for science applications in atmospheric electricity, and are sufficiently simple and compact (eg Figure 2b) to be mounted physically within instruments or indeed even considered disposable. Figure 3 shows examples of surface instruments employing these techniques, mostly constructed at Reading. The Geometrical Displacement and Conduction Current Sensor (GDACCS) (Figure 3a), uses a combination of flat and shaped electrodes to monitor the vertical current density flowing in the global circuit, which is typically $\sim 2$ pA m$^{-2}$ (Bennett and Harrison, 2008). Figure 3b shows a Programmable Ion Mobility Spectrometer (PIMS), which determines positive and negative air ion properties by deflection with an electric field to cause ion flow to a collecting electrode (Aplin and Harrison, 2001).

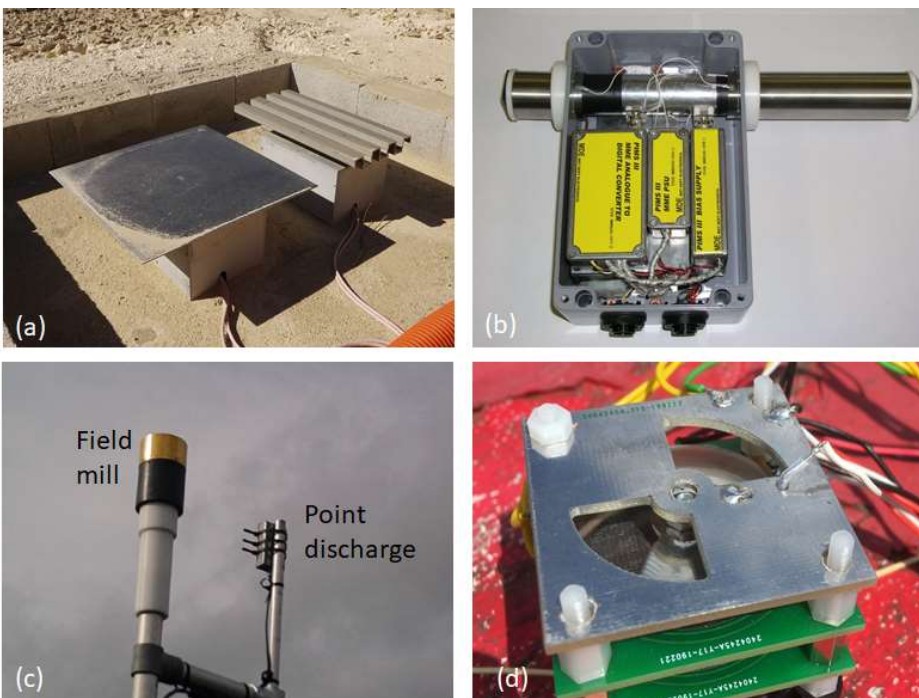

**Figure 3 A selection of atmospheric electricity instruments. (a) Vertical current density sensor (GDACCS), using plate and corrugated electrodes, in the Negev desert. (b) Programmable Ion Mobility Spectrometer (PIMS) to determine electrical properties of air. (c) Electric field mill with sensing surfaces uppermost, and upward pointing point discharge needle (not visible) with logarithmic current amplifier. (d) Miniature field mill with internal calibration electrodes.**

An electrometer voltmeter can measure an electric field by determining the corresponding charge induced in an exposed electrode of known geometry. This offers the possibility of non-contact voltage measurement. A voltmeter operating in this way was first described by Harnwell and Van Voorhis (1933), using a motor to alternately expose and screen the electrode by rotating an earthed shutter. Devices made on this "field mill" differencing principle have been found especially suitable for atmospheric electric field measurements (Lueder, 1943; Mapleson and Whitlock, 1953). Through improvements which removed the brushes earthing the rotating shutter (and therefore reduced the associated wear), field mills have become able to operate continuously in disturbed weather conditions (Chubb, 1990;

1999). Figure 3c shows an upward-pointing commercial field mill of durable design, the JCI131, at Reading University
Atmospheric Observatory. It is mounted alongside a point discharge tip (a fine steel needle) connected to a logarithmic
electrometer which can measure across the wide range of currents encountered in disturbed weather (Marlton et al,
2013). Figure 3d shows a small field mill operating on the brushless principle, developed for balloon use, which can
also generate reference electric fields internally for calibration (Harrison and Marlton, 2020).
**3. Electrical structure of extensive layer clouds**
The explanation for the positive electric potential consistently observed near the surface in fair weather is found
through the global atmospheric electric circuit, originally postulated by C.T.R. Wilson (1921, 1929). The global circuit
allows currents to flow from generating regions (driven by thunderstorms, shower clouds and vertical charge
exchange), to fair weather regions, through which the current passes to complete the circuit. The conduction from
generator regions occurs through the more strongly ionised parts of the atmosphere, and through the earth's surface,
which, compared with the atmosphere, is a relatively good conductor. As indicated above, the concept of the fair
weather branch of the circuit is well-established.

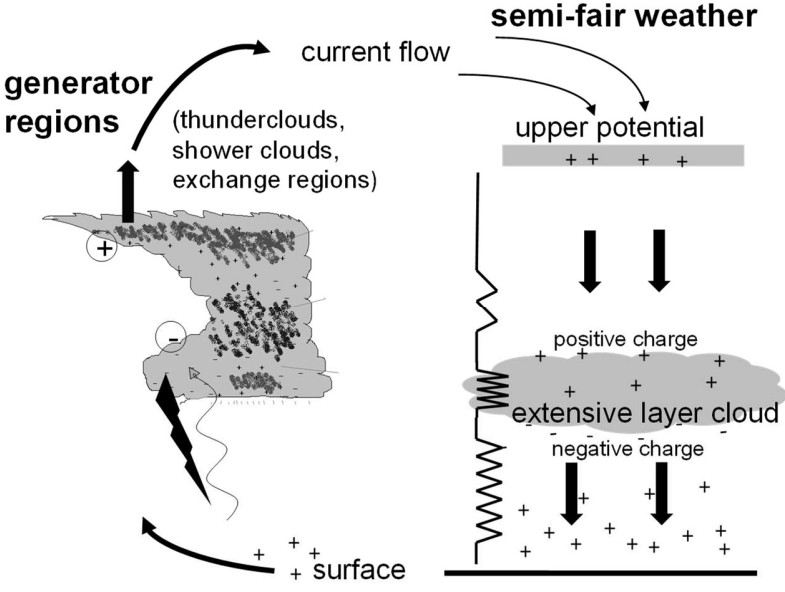

**Figure 4 Conceptual picture of the global atmospheric electric circuit, in which a current is generated in disturbed weather**
**regions, and returns through distant fair weather and semi-fair weather regions. As the current returns through extensive**
**layer clouds, charge accumulates at their upper and lower boundaries.**
Figure 4 summarises current flow in the global circuit, and illustrates the situation which can readily arise when the
fair weather current flowing encounters an electrically quiescent layer cloud. The cloud will present a more resistive
region than the clear air surrounding it, hence the cloud to clear air interface can be understood to provide a transition
in electrical resistance. If the layer cloud is extensive horizontally, the current must pass through it. As it does so, local
space charge is generated at the cloud-clear air boundary, yielding positive charge at the cloud top and negative charge
at cloud base. These circumstances are conveniently referred to as semi-fair weather conditions (Harrison et al, 2020).
Because of the global nature of the current flowing in the circuit, charge at the boundaries of layer clouds is expected
to occur widely. Direct observations of layer cloud charge have, however, rarely been made. Observing just the lower
cloud charge, can, in principle, be achieved by using surface instrumentation such as that of Figure 3c, because of the
influence of the lower charge region on the surface electric field. (This is reminiscent of Canton's approach with an
electroscope, described in section 1.1 above). In persistent and extensive layer clouds, the cloud base charge only
varies slowly, hence fluctuations in the cloud base position can be regarded as representing the motion of a steady
charge, causing changes in the electric field sensed at the surface.

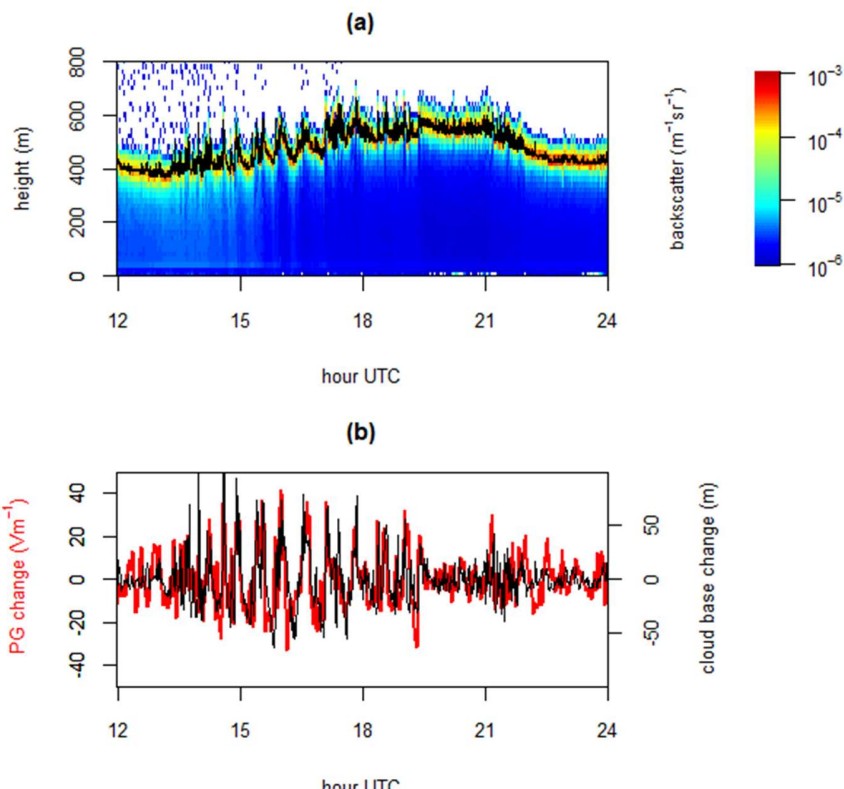


**Figure 5. Measurements from Reading University Atmospheric Observatory on 19th March 2015 showing (a) ceilometer**
**backscatter from cloud base (at 400 to 500m) and (b) fluctuations in the atmospheric electrical Potential Gradient (PG,**
**thick red line) and cloud base height (thin black line).**

Conventionally, the electric Potential Gradient (PG), has been recorded at the surface in fair weather rather than the
electric field[9]. Under a persistent extensive layer cloud, the PG is found to be suppressed when the cloud base height
is lower than about 1000 m (Harrison et al, 2017a). By determining the cloud height using an optical time of flight
measurement, as provided by a laser ceilometer, variations of the cloud base height and PG can be compared. Figure

---

9. The PG is positive in fair weather. Although the electric field has the same magnitude as the PG, it has the opposite

sign; positive charge moves downwards in fair weather.

5 shows an example of a thin (~300m) and low extensive cloud layer, in which the cloud base fluctuations are closely
correlated with the surface PG changes (Harrison et al, 2019). This demonstrates both the existence and persistence
of the lower cloud charge, by a remote sensing approach. Direct measurement within a cloud is needed, however, if
the vertical charge structure is to be examined and quantified.
**4. Radiosondes for atmospheric measurements**
Access to a cloud clearly requires an airborne platform of some kind, other than for the special cases of mountain top
clouds, or surface studies of fog. The transient nature of fogs and the low base typical of extensive layer clouds make
aircraft flight plans difficult, which are required well in advance. An alternative is to take advantage of the standard
meteorological method of sending instrument packages - radiosondes - aloft by weather balloons (Figure 6a), and
return the measurements by radio. These devices, originally known as radiometeorographs, were developed in the
1920s to replace mechanical recording devices (e.g. Idrac and Bureau, 1927) and rapidly found widespread use
(Wenstrom, 1934). Commercial devices followed, notably developed by Vilho Väisälä (Väisälä, 1932), whose name
is carried by the Finnish company he founded.
Radiosondes have a well-established global role in obtaining routine meteorological data, and can, at some sites at
least, be launched rapidly in response to conditions under well-established regulations. However, standard radiosonde
systems typically only measure the conventional thermodynamic variables of meteorology (atmospheric pressure, air
temperature and relative humidity, or "PTU") and their immediate location, from which the wind vector can be
inferred. For measurements beyond these, additional sensors and data transfer systems will be required.

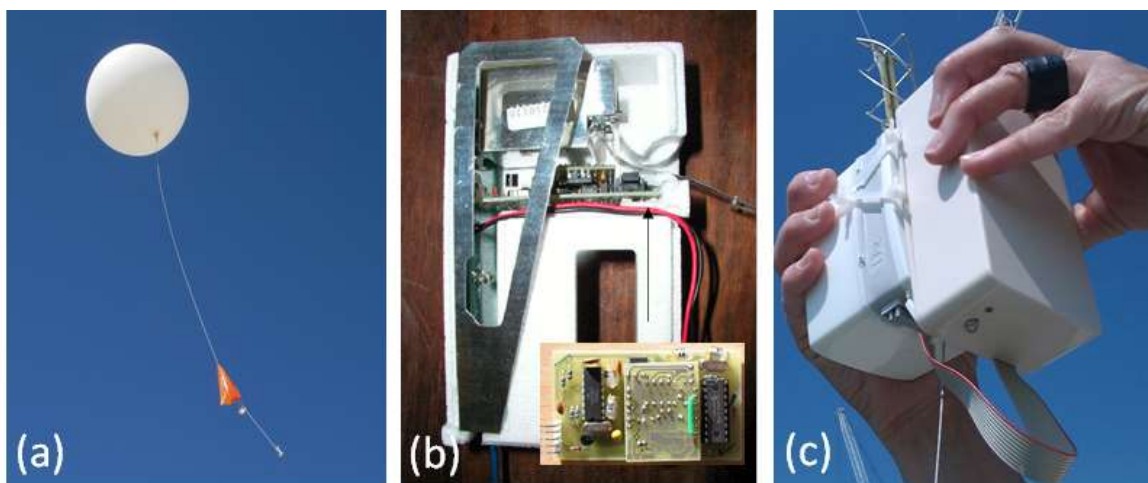


**Figure 6 (a) Radiosonde in flight, showing the carrier balloon and parachute for the descent. (b) RS80 radiosonde partially**
**disassembled, showing the data acquisition system (RS80DAS) mounted within an internal cavity. (Inset photograph shows**
**the RS80DAS circuit board). (c) PANDORA system fixed to a RS92 radiosonde immediately pre-launch, showing the ribbon**
**cable data connection to the radiosonde ozone interface connector.**

**4.1 Research radiosondes**

Radiosondes have long provided research measurements not used directly in weather forecasting, also referred to as "operational meteorology". An example is the ozonesonde (Brewer and Milford, 1960), which carries electrochemical apparatus to determine ozone concentration in the stratosphere. Sensors for radioactivity (Koenigsfeld, 1958), turbulence (Anderson, 1957), cosmic rays (Pickering, 1943), aerosol properties (Rosen and Kjnome, 1991) and supercooled liquid water (Hill and Woffinden, 1980) have all been carried by radiosondes, and there are many more examples. Research radiosondes have also been used in atmospheric electricity, such as for charge measurements in thunderstorms (Takahashi, 1965), but also for PG, conductivity and conduction current density in semi fair weather conditions (eg Venkiteshwaran,1953; Jones et al, 1959; Olson, 1971; Gringel et al, 1978). Establishing the vertical thunderstorm charge structure by a balloon-carried recording instrument, the "alti-electrograph" (Simpson and Scrase, 1937) was especially important in the acceptance of the global circuit concept (Simpson, 1949). The long time series of cosmic ray measurements made by the Lebedev institute (Stozhkov et al, 2009) exists due to regular weekly launches of balloon-carried instruments from several sites.

Reviewing previous approaches illustrates the range of different technologies which have been used, either by adapting existing meteorological devices or, in some cases (e.g. the Lebedev instruments), developing a custom radiosonde. A disadvantage of adaptation is that one or more channels of meteorological data may be lost to in providing telemetry bandwidth for a new quantity. For applications which need the meteorological data, this is clearly undesirable. Instead, if the routine radiosondes used in operational meteorology are harnessed to carry additional sensors without losing their core meteorological data, a much greater opportunity for new measurements presents itself, allowing access to the existing global launch and reception infrastructure of more than 1000 soundings daily. This has led to a new strategy of making "piggy-back" systems which provide additional measurement capability on standard meteorological radiosondes, whilst preserving the existing meteorological data. Furthermore, if the add-on devices are made straightforward to use, more launch opportunities can be obtained worldwide from those familiar with using meteorological radiosondes routinely, but who are not specialists in the research quantity sought.

The associated programme of work at Reading has mostly built on the Vaisala range of radiosondes, largely because the related equipment was already available at the University. Many other commercial radiosondes are available internationally, and the principles developed in using a programmable interface to support a range of sensors and communicate with the radiosonde are very flexible, and amenable to other commercial systems too.

**4.2 Interfacing and research data telemetry**

Since the late 1990s, Vaisala has manufactured three major radiosonde versions, the RS80, the RS92 and the RS41. Table 1 provides a summary of how additional measurements have been obtained from each model without

compromising the standard meteorological data. Systems added will encounter a wide operating temperature range, as even the polystyrene-insulated internal environment of a RS80 radiosonde can drop to at least -55°C, (Harrison, 2005). Connections and mountings need to be robustly implemented, as accelerations of ±60 ms$^{-2}$ associated with vigorous turbulence can occur (Marlton et al, 2015). There are also power constraints to consider, in that the radiosonde battery must not be excessively drained or all subsequent data transmission will be lost. The typical duration of an operational radiosonde ascent is one hour, followed by slightly quicker descent by parachute. For each of the systems in Table 1, an operating time of 3 hours was typically achieved, either by minimising the sensor current drawn (RS80 and RS92), or by including supplementary batteries (RS41). Further, the free lift of the standard balloon must not be exceeded if the equipment is ever to leave the ground. Cost, as the systems are only very rarely recovered after a flight, should also be minimised.

**Table 1 Vaisala meteorological radiosondes and their use with additional research sensors**

| Radiosonde model | Radiosonde battery | Meteorological data telemetry | Details of additional research data transfer | data transfer | Reference |
|---|---|---|---|---|---|
| RS80 | 18V (wet cells) | switched analogue tones | analogue-modulated frequency on 100k Hz LORAN channel | single analogue channel | Harrison (2001) |
| | | | RS80DAS digital interface using Bell 202 modem tones at 300 baud, injected into radiosonde audio; 4 mA consumption; mass 16 g; four 12bit channels; +18V and +5V supplies; | 10 x 12bit samples per second | Harrison (2005a) |
| RS92 | 9V (6x AA alkaline) | digital | PANDORA interface system using radiosonde ozone interface; 3mA consumption; mass 110 g boxed; 16bit and 10bit channels; +16V, ±8V and +5V supplies; | 4 x 16bit samples per second | Harrison et al (2012) |
| RS41 | 3V (2x AA lithium) | digital | PANDORA4 interface system using radiosonde "special sensors" serial data transfer | up to 200 bytes per second | Radiosonde: Vaisala (2020) |

The first experiments were with the analogue RS80 radiosonde. The RS80 used a sequence of audio tones to send its PTU measurements, and it also provided an additional channel to relay LORAN (a LOng RAnge Navigation system using very low frequency) positioning signals, later entirely superseded by satellite GPS (Global Positioning System). As LORAN was not implemented in the Reading Meteorology Department's radiosonde system, this offered a spare

channel to send additional data, through an analogue voltage-to-frequency converter varying within a few percent of 100 kHz. This signal was recovered as a voltage, by passing the modulated 100 kHz signal to a phase-locked loop (PLL), and the tracking voltage logged with a 12bit analogue to digital converter. Time stamping of the radiosonde data and PLL data on separate logging computers allowed the two different files to be aligned. Regular switching to full scale at the radiosonde end was also applied to allow correction for non-linearity and thermal drift.

Limitations from single channel analogue transfer led to a more extensive digital data acquisition system (RS80DAS), fitted in a cavity within the RS80 (Figure 6b). The modem used was chosen for signalling tones (1200/2200 Hz) outside the audio frequencies already used for the RS80's meteorological data[10]. Samples from four 16bit ADC channels were formatted by a microcontroller and sent to the modem for onward transmission, decoded by a further modem at the receiver to yield a serial data stream. This arrangement provided a much more temperature stable system (15 mV error in 5 V full scale across a 60 °C temperature change), and, importantly, could convey simultaneous data from multiple sensors (Harrison, 2005a).

When a subsequent radiosonde design, the RS92, was introduced, its smaller dimensions meant that it was no longer possible to fit the existing data acquisition interface within the radiosonde. The RS92 was digital, with special provision for sending additional data when deployed as an ozonesonde. A new data acquisition system was designed to generate a data stream which mimicked that expected from the sensor in the ozone application. For this, the data was buffered in bursts at the rapid rate required by the radiosonde, using a first-in-first-out (FIFO) shift register. The complete system was called PANDORA (for *P*rogammable *AN*alogue and *D*igital *O*perational *R*adiosonde *A*ccessory)[11], which was physically attached to the RS92 radiosonde using cable ties (Figure 6c). It provided four 16bit and two 10bit analogue channels, and regulated power supplies for the sensors connected. The radiosonde's meteorological data was shown to be unaffected by the PANDORA's addition (Harrison et al, 2012).

With more soundings undertaken for an increasing range of different scientific objectives, the inherent versatility became time consuming to implement, as, for each custom system constructed for a particular application, individual wiring and software was required. A much more adaptable system, PANDORA3, was devised, based on stackable sensor boards mounted above each other (Figure 7a), with a consistent physical form and arrangement of connectors on each of the sensor boards. Each sensor board carries its own microcontroller (or microcontrollers), which only returns data to the PANDORA3 when polled with its specific address. This allows the PANDORA3 to configure itself and format its data stream automatically for whatever combination of sensors is fitted.

---

10 Some agencies and individuals monitor radiosonde transmissions in their nearby airspace. Changes from the routine radiosonde data transmission formats may attract additional attention.
11 The acronym PANDORA also discourages opening of the box, and other tampering, so has been retained.

359 As the radiosonde technology has evolved to become more compact, their battery voltage and spare battery capacity

360 has reduced. The PANDORA4 now carries three AAA cells (Figure 7b) to power itself and associated sensors, and

361 includes supply voltage converters to maintain compatibility with earlier PANDORA systems.

362

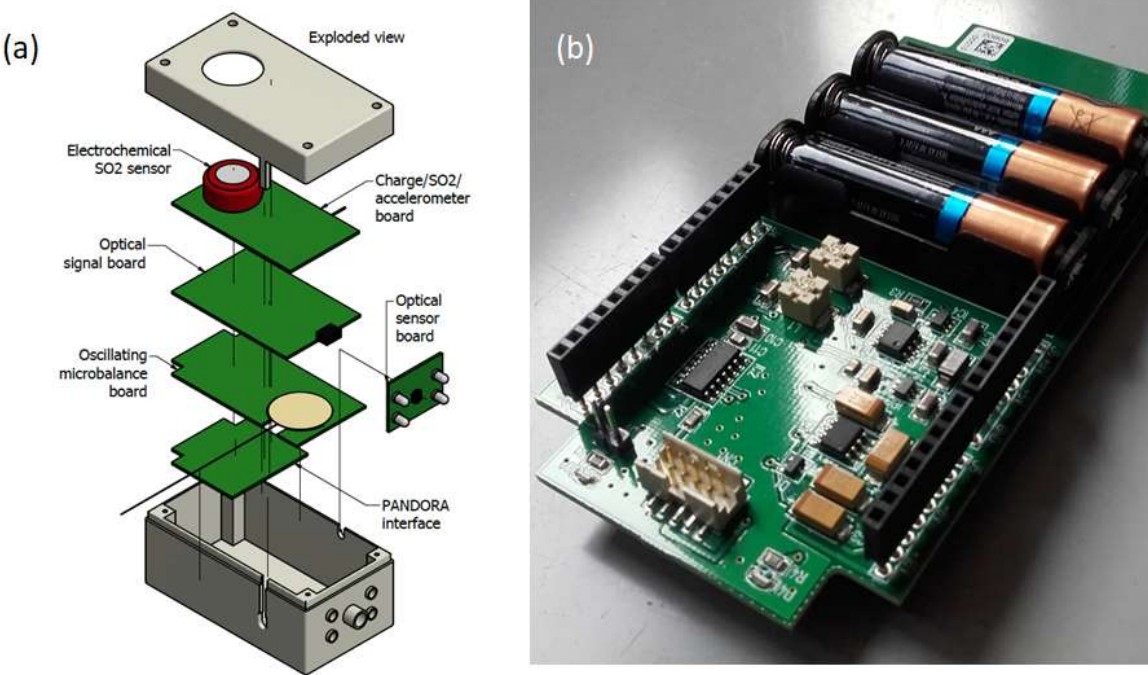

363

**Figure 7 The PANDORA4 system for support of additional radiosonde sensors. (a) Stackable arrangement of multiple sensor boards, in this example including an accelerometer, collecting wire for an oscillating microbalance, cloud, charge and gas detectors. (b) PANDORA4 data board for use with RS41 radiosonde, showing stackable connectors and additional battery supply.**

Some of the sensors devised for various atmospheric measurements, motivated originally by the cloud charge
application, are now described.

**4.3 Electrometer radiosondes**
Measurement of atmospheric charge using a radiosonde requires a sensing electrode and electrometer able to measure
the charge collected or induced, with some sort of data telemetry as described above. The electric potential of the
radiosonde changes as it rises through the atmosphere, but more slowly than that of a small sensing electrode, causing
a current to flow transiently which can be measured. One electrode configuration which has some simplicity, suitable
for large electric fields, is a point electrode for corona discharge. Figure 8a shows a corona sonde from 1998, in which
a needle electrode was connected to a current amplifier, following the electronic principles of section 2.2. It was not,
however, a convenient arrangement to launch, not least because of the proximity of the sharp needle to an inflated
rubber balloon. Rounded electrodes are preferable, with, the connection between the electrode and the electrometer as
short as possible to reduce leakage. A novel capsule well suited to this application was found within a "Kinder Egg",
housing a self-assembly toy contained within a confectionery egg. This capsule is manufactured from hydrophobic
material, is strong enough to resist modest impacts, and is water-tight, offering some protection to any electronics
mounted within it.

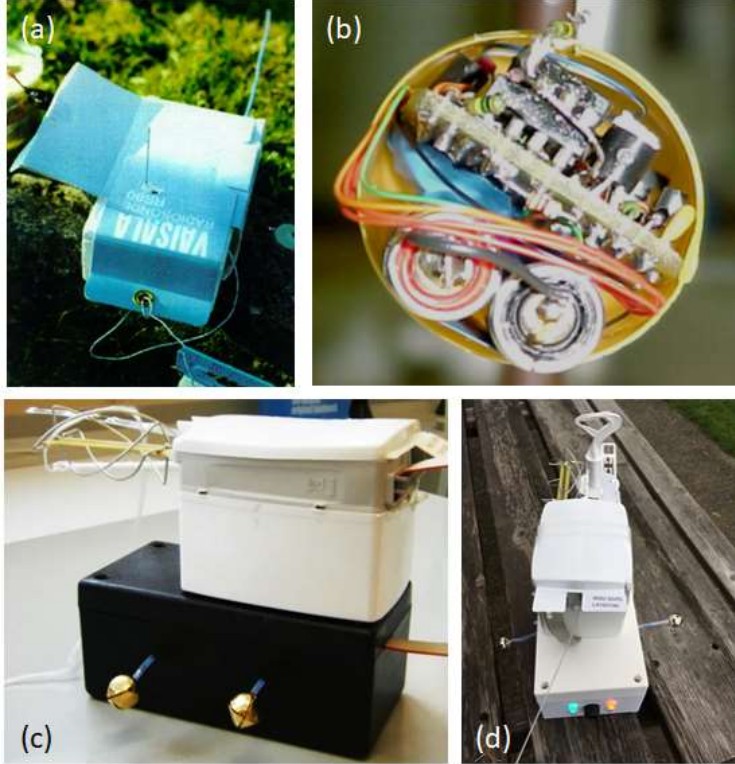


**Figure 8 Radiosonde electrometer sensors. (a) Corona discharge needle, protruding from a RS80 radiosonde. (b) Extended range electrometer mounted within a "Kinder Egg" housing with a conductive coating applied. (c) Double electrode electrometer with different linear ranges. (d) Electrode pair with linear and logarithmic electrometers, and multi-wavelength cloud sensor.**


Figure 8b shows the electrometer circuitry contained with a Kinder Egg capsule (Harrison, 2001)[12]. A graphite
conductive coating was painted on, connected electrically with silver-loaded epoxy adhesive. The wide range
electrometer was powered from car key fob batteries (giving 24 V), with the system electrically isolated and an optical
connection made to the radiosonde. Because of the risk that the sensor could become irrecoverably electrically

---

12 Many such capsules were used in atmospheric experiments, including 100 in a single project for the US Navy. Leaving the Kinder Eggs in the Meteorology Department coffee room was found extremely effective in distributing the initial dismantling task. This inspired footnote 7 to the 2001 paper, "The outer chocolate and foil coating must first be removed", which was the first mention of chocolate as a substance since the *Review of Scientific Instruments'* foundation in 1930.

saturated early in a flight, regular reset switching was included as a precaution against data loss. A semiconductor
switch with suitably negligible leakage (<10 fA) was developed specially, using the gate-source connection of a JFET
as a diode, across which the voltage difference was kept around 1 mV when off. This was switched on for 1 s every
10 s, also providing a regular full-scale value with which to correct for the drift inherent in the analogue telemetry.
Figure 8c shows a later version of the charge detector, using smaller spherical electrodes, again a mass-produced item,
originally intended as a small (6 mm diameter) pressed metal bell. Initially, a voltmeter follower circuit was used,
with reset switching as previously (Nicoll and Harrison, 2009), and the charge calculated from the capacitance.
However, the smaller size led to more difficulties with saturation. It was found more satisfactory to measure the current
flowing, arising from changes in the electrical environment through which the sensor passed (Nicoll, 2013). In Figure
8c, two sensors were connected to electrometers with different ranges, to allow different cloud charges to be
investigated, as the optimum range had to be established empirically. Figure 8d shows a further system which
combines linear and logarithmic response electrometers (Harrison et al, 2017b), to provide accuracy in one case and
wide dynamic range in the other.

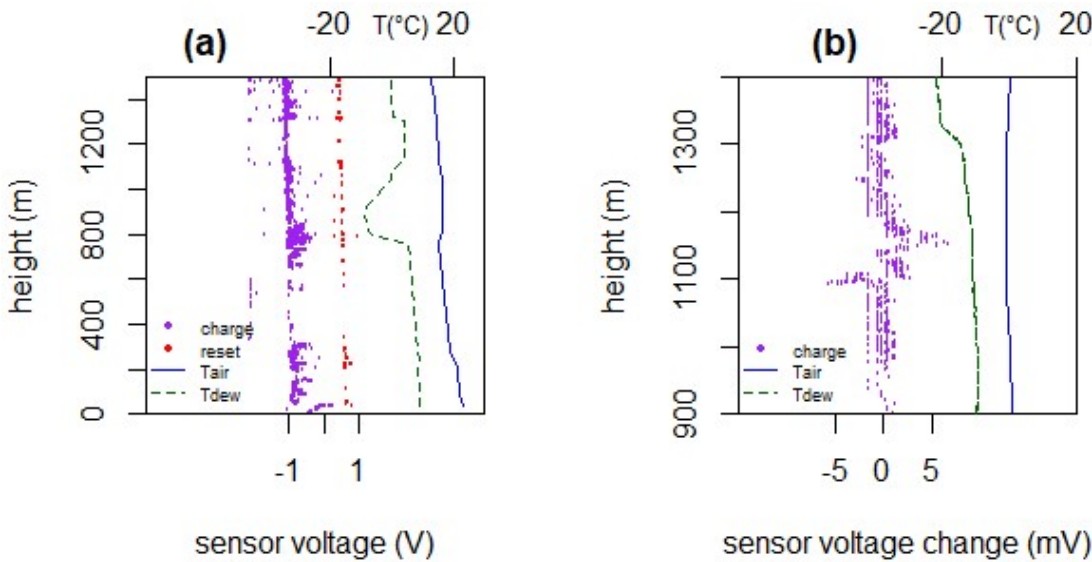


**Figure 9 Vertical soundings from Reading on (a) 24th August 2000 and (b) 3rd March 2004. Points show the recovered**
**sensor voltage (scale on lower horizontal axis), with full-scale pulses highlighted in red on (a). Solid blue lines and dashed**
**dark green lines show the air temperature and dewpoint temperature (temperature scale on upper horizontal axis).**

Figure 9 shows examples of soundings made with the early Kinder Egg sensors. Figure 9a was obtained using the
analogue system, and the regular reset switching is apparent. A response in the charge sensor is apparent at about
800m altitude, probably associated with the top of the atmospheric boundary layer. (Similar fluctuations can provide
detailed information on the electrical structure throughout this atmospheric region, Nicoll et al, 2018). Figure 9b shows
an example of the complex charge structure commonly observed, although in this case, at the limit of the measurement
resolution of the digital system. Both soundings, however, show that the vertical resolution of the meteorological data
is relatively coarse when compared with the vertical detail in the electrical structure. Unfortunately, it was concluded
that the meteorological data alone is inadequate for interpretation of the electrical measurements, and that other
simultaneous observations would be needed. A series of further sensors has accordingly been developed, to take
advantage of the additional data telemetry available in the digital data acquisition system. Above all, the most
important additional requirement for determining cloud charge has been an independent method for reliable cloud
identification. Other corroborating sensors, for example reporting the motion of the radiosonde, can bring value by
allowing the sensor orientation to be monitored. Development of these additional sensors measuring other quantities
beyond charge measurement is now addressed[13].

## 4.4 Optical cloud detection

Cloud is conventionally determined on a radiosonde sounding by using humidity information, typically obtained by a
capacitive relative humidity sensor. These sensors have a finite response time, which, when combined with the ascent
speed of the radiosonde, prescribes a minimum thickness of cloud which can be detected. If the sensor time response
is 10 s, and the ascent speed 5 m s$^{-1}$, this thickness would be of order 50 m. As the soundings of Figure 9 show,
atmospheric charge layers can be much thinner than this, so the humidity method is clearly insufficient. An optical
method can be expected to have a much more rapid time response, for example using a photodiode as a detector of
optical changes caused by cloud.

Two approaches for optical cloud detection have been investigated. The first method was passive, in that the
photodiode was used to detect cloud-induced changes in received solar radiation, and the second active, using a bright
local source of illumination to generate backscattered light when droplets are present. The first method can only work
in daylight, and the second method, initially at least, was intended to be complementary, for use at night.

In the passive cloud detector arrangement, an inexpensive and commonly available VT8440B photodiode (peak
spectral response at 580 nm) was implemented with an amplifier circuit, to provide a large signal input to the data
acquisition system (Nicoll and Harrison, 2012). This essentially provided a measurement of solar radiation, falling
either directly on the photodiode, or as diffuse solar radiation after scattering of sunlight in cloud. The presence or
absence of cloud modifies the variability in solar radiation. Within a cloud, the light is essentially isotropic, so
swinging motion of the radiosonde has little or no effect, but away from cloud, the light intensity varies strongly with
direction. Figure 10a shows the change in solar radiation received by a downwards-facing photodiode as it rises
through low stratiform cloud. At the cloud base, the radiation begins to increase steadily with height as the optical
thickness of the cloud diminishes. As the instrument emerges from the cloud top, the solar radiation variability sharply
increases, due to swing of the radiosonde beneath the carrier balloon. The relative humidity sensor data is provided

---

13 Extending the range of sensors, although apparently moving away from the initial science objective, has also
brought the benefit of diversifying funding sources.

for comparison, in which much less distinct boundaries are apparent at the cloud base and cloud top. With the relative
humidity measurement alone, the cloud position would only be poorly defined.

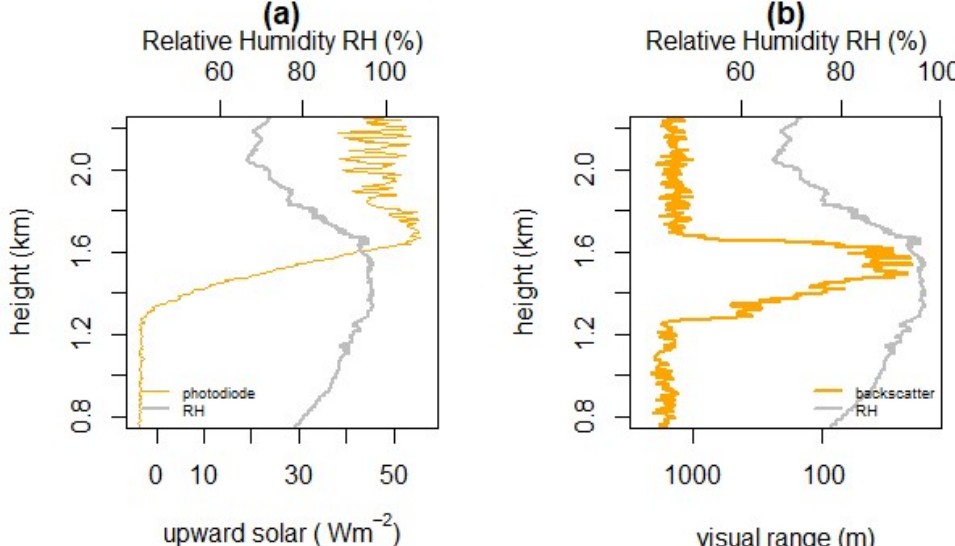

**Figure 10 Sounding of a cloud layer from Reading on 22nd April 2014. (a) Solar radiation from a downward pointing**
**photodiode was recorded (thin orange line) simultaneously with (b) the received backscattered light from ultrabright yellow**
**light emitting diodes (thick orange line), expressed in terms of equivalent visual range. The Relative Humidity profile (thick**
**grey line) is shown on both plots. (Adapted from Harrison and Nicoll, 2014).**
The active cloud detection method requires strong local illumination, ideally from an open source to reduce difficulties
with ice formation. In aerosol sensors, laser sources are used within chambers into which air is pumped, but an open
laser source is unlikely to be safe on balloons because the direction would be variable and uncontrolled. For this
application, high intensity light emitting diodes (LEDs) are ideal, as they are compact and highly efficient. Even so,
the light scattered by water droplets in clouds generates only a small signal and substantial amplification is required.
If the LED light source is modulated, the noise inherent in this process can be reduced. In addition, since modulation
provides a varying signal which contrasts with the steady signal of sunlight, the modulated signal can be amplified
selectively despite strong sunlight[14]. By designing the first amplifier stage with a small gain to allow a wide dynamic
range, adding high pass filtering, further amplification and phase-sensitive detection, detection of the backscattered
light even in daytime conditions becomes possible, as demonstrated in Harrison and Nicoll (2014). Figure 10b shows
the active cloud detection method operating during the same ascent as for Figure 10a. With the active method, the
cloud base and cloud top are both very sharply defined, consistently with that expected from the passive detector
shown in Figure 10a.

---

14 To achieve this, the photocurrent in sunlight must not saturate the first amplifier stage.

These optical sensing methods allow cloud boundaries to be determined to much greater resolution than by the standard relative humidity sensor of a radiosonde, and typically to better than ±10m. Consequently, by using these optical methods, some thin layers of cloud may become detectable which would not be registered by the standard approach using a relative humidity sensor.

**4.5 Turbulent motion**

When a radiosonde is launched, it is common to see the ascending instrument package swinging or twisting. Through collaborating on an arts project[15], in which a camera transmitted images continually from a rising balloon package, it was clear that the irregular motions continued throughout the ascent and were not just associated with the launch. The motion of a radiosonde package is complex, as it responds to both the atmospheric motions encountered by the carrier balloon and a pendulum motion from the combination of the instrument package and the attachment cord.

A first attempt to monitor the radiosonde's motion was through including a small sensor for the terrestrial magnetic field, the signal from which would only vary if the instrument package was in motion. This was, essentially, a compass for a radiosonde, using a Hall effect sensor (Harrison and Hogan, 2006). With this magnetometer-sonde, multiple soundings within hours of each other showed consistent magnetometer variability in the same region of the atmosphere, implying a turbulent atmospheric region (Harrison et al, 2007). Following a suggestion that the vertical magnetic variability would prove most useful (Lorenz, 2007), the three orthogonal components of the magnetic field were measured simultaneously, and rapidly, and processed on the radiosonde to conserve the amount of data sent over the radio link. Through this work, the vertical component was indeed found to be the most successful, and the response of this component from the magnetometer-sonde in the lower atmosphere could be calibrated against lidar determination of the turbulence through which it passed (Harrison et al, 2009).

Later developments in semiconductors have allowed miniature accelerometers to be used instead of the Hall sensor, which directly sense the forces on the radiosonde. With these it was found possible to calibrate the motion of the radiosonde to provide the Eddy Dissipation Rate, a measure of atmospheric turbulence (Marlton et al, 2015).

Lorenz (2007) also commented on the relevance of the platform motion work to planetary exploration. In a later paper, Lorenz et al (2007) reported motions of the Huygens probe as it descended through the atmosphere of Saturn's moon Titan. The power spectrum of these motions showed good agreement with that found within turbulent terrestrial clouds by Harrison and Hogan (2006), supporting arguments for turbulence within the methane clouds of Titan.

---

15 "30km" was produced by Simon Faithfull (https://www.fvu.co.uk/projects/30km ).

## 4.6 Ionisation and radioactivity

Generation of small ions in the atmosphere leads to the finite electrical conductivity of air, from which current flow in the global circuit results. In conductive air, charge on droplets and particles does not persist for long. Measuring the conductivity or the ionisation is therefore an aspect of characterising the background electrical environment.

Conductivity is conventionally measured using a "Gerdien tube", which consists of a rod electrode, centred within an outer coaxial cylindrical electrode (e.g. Aplin and Harrison, 2000, see also Figure 3b). For a given voltage applied across the electrodes, the current flowing between the electrodes is proportional to the conductivity. The original method used by Gerdien was to determine the rate of decay of charge on the central electrode (Gerdien, 1905). A similar approach was developed for radiosonde use (Nicoll and Harrison, 2008). However, maintaining good insulation within a droplet-laden environment proved difficult or impossible, and it became clear why there are few reliable measurements of in-cloud conductivity.

A practical alternative is to measure the ion production rate at the height of the radiosonde, using a Geiger tube sensor, in a "Geigersonde". The tube is triggered by energetic particles primarily from incoming cosmic rays, and the ionisation rate in the nearby air can be calculated from the count rate. To operate a Geiger tube a high tension (HT) bias is needed (300 to 500 V), and a counting device which can be read at known intervals. The dimensions of the tube determine the sampling volume and the count rate. As mentioned, this approach has been used on many research radiosondes, and, most notably, supported the original indication of a maximum in ionisation - the Regener-Pfotzer (RP) maximum - in the lower stratosphere (Regener and Pfotzer, 1935).

For modern meteorological radiosondes only a relatively small Geiger tube can be carried, with supporting electronics. The Neon-Halogen filled LND714 Geiger tube (22mm long x 5mm diameter, mass 0.8g) has now been used in many flights. Although the count rates are relatively small - around 60 counts per minute at the RP maximum - using a pair of tubes allows some averaging and determination of variability as well as adding confidence if the two count rates are similar. Further, coincident triggering of the two tubes can be used to detect particles which are sufficiently energetic to pass through both tubes (Aplin and Harrison, 2010), although, ideally, the orientation of the tubes should also be monitored. In the Reading Geigersonde implementation (Harrison 2005b; Harrison et al, 2014), the HT supply is generated using voltage multiplication to avoid the weight of a transformer and separate on-board digital counters are triggered by the Geiger pulses. The counters are interrogated by the PANDORA interface, every 30 s, together with the HT voltage. Separating the counting and data transfer hardware ensures that pulses are not missed during the data transfer to the PANDORA.

Although the greatest particle flux is generated in the stratosphere, at the RP maximum, some of the secondary particles formed by decay of the primary particles from space reach the surface. Of these, the greatest flux at the surface is that of neutrons. A global network of Neutron Monitors (NM) provides continuous monitoring of particles entering the atmosphere. Figure 11a shows the variations obtained by the NM in Oulu, Finland, around the cosmic ray

maximum associated with the solar minimum of 2008. Occasional Geigersondes were launched from Reading and
other sites in the latter part of this period. These sounding times are marked on Figure 11a, with the vertical count rate
profiles obtained shown in Figure 11b. By plotting the count rates obtained at the RP maximum against the NM data
at the same time, a positive correlation emerges (Figure 11c), demonstrating the relationship between energetic
particles at the surface and ionisation in the lower stratosphere. There is a much poorer, or absent relationship between
NM data and ionisation in the lower troposphere, due to the terrestrial ionisation sources present and the differences
between generation of neutrons and the other ionising secondary particles.

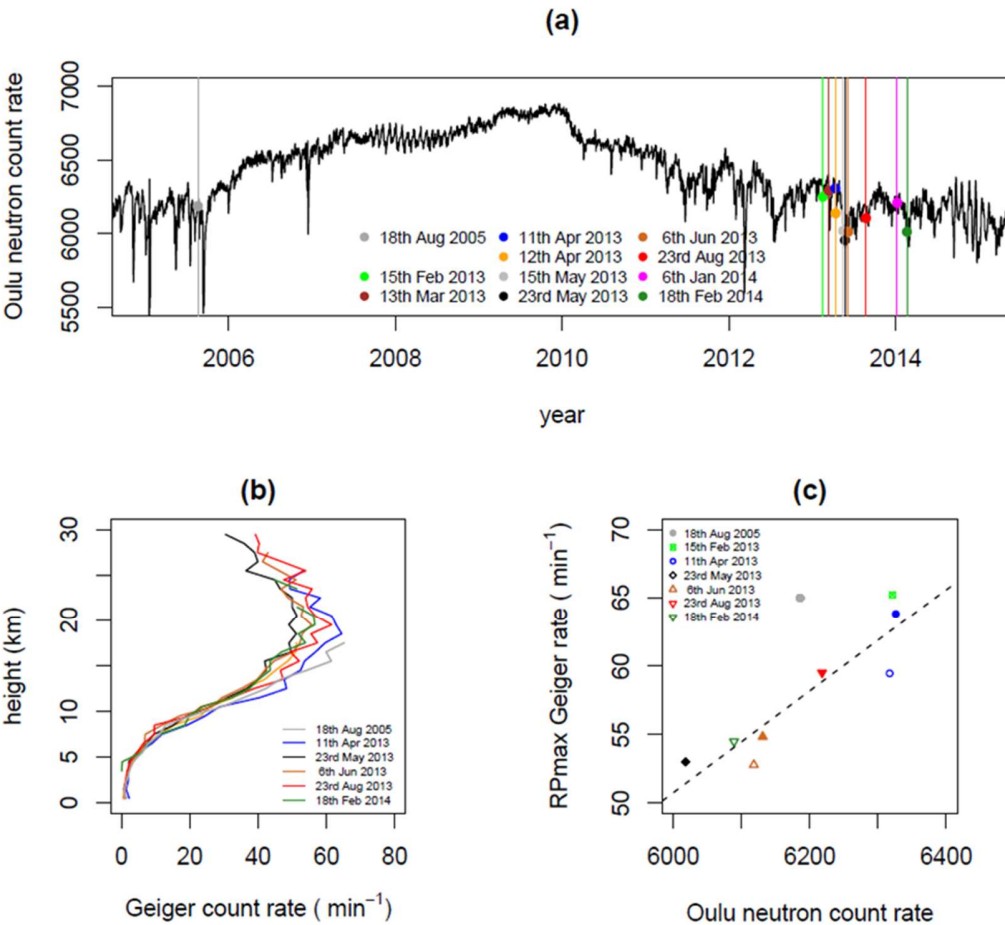


**Figure 11 Comparison of Geigersonde profiles with the surface Neutron Monitor at Oulu. (a) Oulu neutron data time series,**
**with Geigersonde flights from Reading marked. (b) Count rate profiles from selected Geigersonde soundings, showing the**
**increase from the surface to the Regner-Pfotzer (RP) maximum at around 20 km altitude. (c) Comparison of the RP**
**maximum Geiger count rate (RPmax) with the Oulu neutron monitor count rate at the same time (solid symbols ascent**
**data, and hollow symbols descent). (Adapted from Harrison et al, 2014).**
The Geigersonde sounding in Figure 11b with the greatest ionisation at the maximum was in fact associated with a
strong solar flare, on 11th April 2013 (Nicoll and Harrison, 2014). This sounding was made opportunistically in
response to the flare, with the balloon flight around the maximum of the increase in solar energetic particles, followed
by a second flight the day after (Figure 12a). Above about 9 km, the count rates of the Geigersonde were much greater

than those typically found, suggesting an increased flux of ionising particles (Figure 12b). In addition, the coincidence rate between the two Geiger tubes, which is a measure of the abundance of energetic particles, also greatly increased (Figure 12c). The balloon burst about an hour after the launch, and the Geigersonde descent encountered reduced, although still exceptional, count and coincidence rates. The sounding made the following day was unremarkable in comparison. Considering again the proton flux variations in Figure 12a, the lower energy (10 MeV and 30 MeV) protons were still increasing at the time of the flight on the 11th April, but the higher energy (>60 MeV) protons had become steady, implying that the increased coincidence rate was related to higher energy particles. Using the temperature profiles obtained from the meteorological sensors, an increase in count rates on 11th April 2013 can also be seen to have occurred in the upper troposphere.

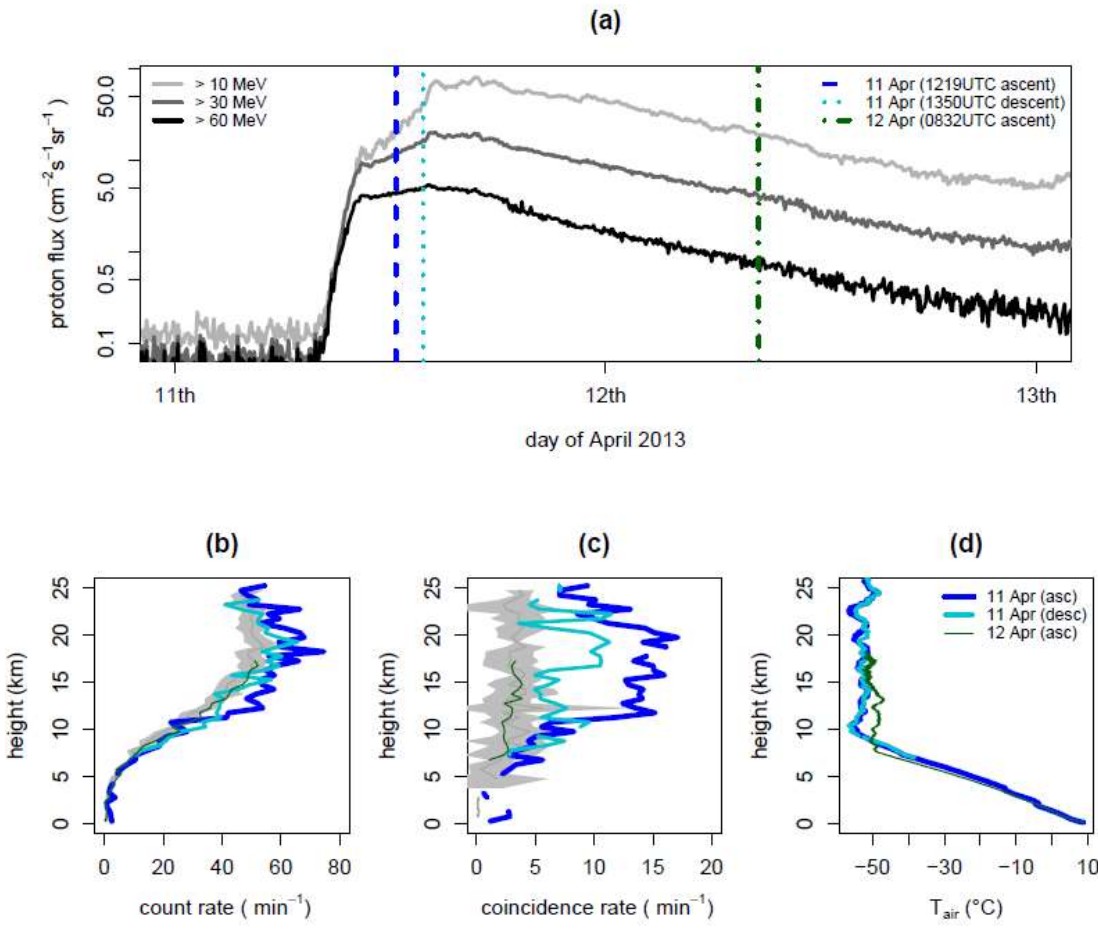

**Figure 12 Geigersonde flights during the solar flare of 11th April 2013. (a) Proton flux time series from satellite (GOES-13) detectors, for proton energies greater than 10 MeV (light grey line), 30 MeV (dark grey line) and 60 MeV (black line). Vertical lines mark Geigersonde launch times from Reading for 11th and 12th April (dashed and dash-dotted), and the burst time beginning the descent on 11th April (dotted). Soundings of (b) count rate and (c) tube coincidence rate from ascent and descent on 11th April (thick and medium lines), and ascent on 12th April (thin line). Grey bands show confidence range (2 standard errors) from undisturbed flights. (d) Air temperature profiles on 11th and 12th April. (Adapted from Nicoll and Harrison, 2014).**

576

**4.7 Dusts and volcanic ash**

As well as droplets, charging of dust occurs in the lower atmosphere, which is highly likely to be a characteristic of other planetary atmospheres too (Harrison et al, 2016). Radiosondes instrumented with charge sensors provide a means of observing this. Including an optical aerosol counter allows the properties of the dust to be determined simultaneously. Such instrument packages have been used to sample Saharan dust aloft in Cape Verde (Nicoll et al, 2011), and during the national emergency associated with the volcanic eruption plume from Eyjafjallajökull in 2010 (Harrison et al, 2010). In both cases enhanced charging was observed in regions of increased particle concentrations. The Eyjafjallajökull plume measurements were made following an urgent request from the UK Government, for which a special expedition was undertaken (Figure 13a, b), using the devices designed for the work in Cape Verde (Figure 13c). The sounding demonstrated the presence of small particles aloft, which was not associated with cloud (Figure 13d, e). Due to the haste[16], the charge sensors used in Cape Verde were not removed. This was fortuitous, as it allowed charge in the plume to be observed (Figure 13f), which, given the distance from the eruption in Iceland, would have been generated during the atmospheric transport of the plume.

---

16 Some brief recollections are given in Harrison (2021). The value of these radiosondes in locating the plume was reported to the UK Parliament's Science and Technology Committee (3rd November 2010).

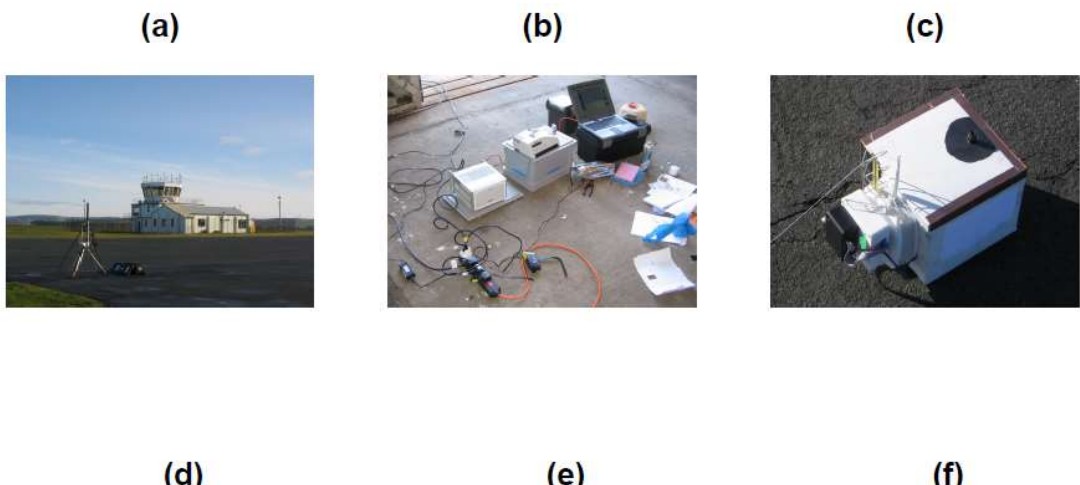

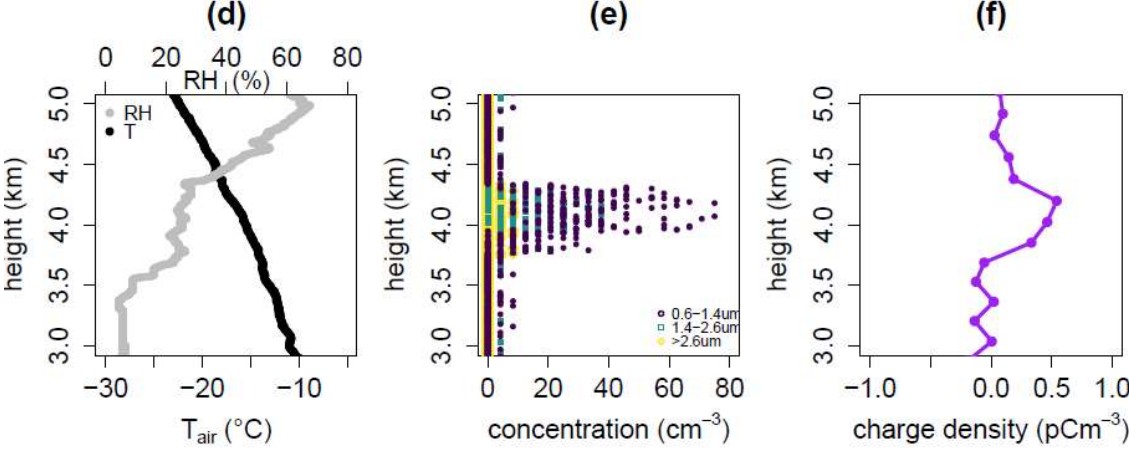

**Figure 13 Research radiosondes used in an emergency to locate the 2010 Eyjafjallajökull volcanic plume over the UK. (a) (b) Temporary receiving station at RAF West Freugh near Stranraer, Scotland, under clear skies, for launch of an enhanced sonde able to measure particle size distribution. (d) Part of the meteorological sounding following launch at 0830UTC on 19th April 2010. (e) Profile of plume concentrations for micron diameter particles and (f) simultaneous measurement of charge density within the plume region.**

The hazard to aircraft of the volcanic ash is directly related to the mass concentration of the particles present, due to deposits in aircraft engines. Although estimates can be made from satellites, information on the optical properties of the ash, which vary with its composition, are also needed which may not be immediately available. An alternative approach for in situ sensing is to collect the ash and determine the mass concentration directly. One method is to use a vibrating rod method (or "oscillating microbalance"), as also used for supercooled water collection. As the mass accreted on the rod increases, its natural oscillation frequency decreases. With accurate frequency measurements and knowledge of the collection efficiency, the concentration encountered can be found. A radiosonde system for this has been developed (Airey et al, 2017), which combines hardware (phase-locked loop) and software (the Fast Hartley Transform) approaches for determining the free oscillation frequency. Droplet collection experiments in the Arctic have shown agreement with another vibrating rod system collecting supercooled liquid water (Dexheimer et al, 2013). In the ash collection mode, adhesive is first applied to the vibrating rod. Figure 14 shows the effect of introducing pumice into a region monitored by the optical cloud sensor, also allowing collection by the rod of the oscillating

microbalance. Clearly, physical collection will require more material than for optical detection, but, as impaction is
the process which presents the hazard to aircraft engines, the application to airspace security is much more direct.

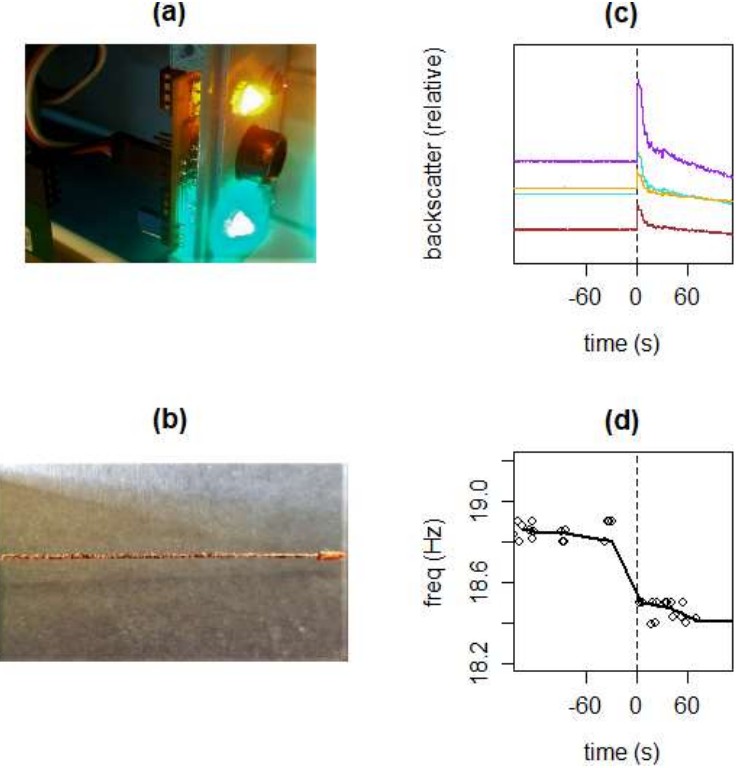

**Figure 14 Comparison of (a) optical detector and (b) oscillating wire ash collector, shown after the collecting wire became**
**coated with pumice. (c) shows data from the optical detector's four channels (relative responses), during which pumice**
**was introduced, at time 0 s. (d) shows the simultaneous change in vibration frequency of the adhesive-coated collecting**
**wire, as the pumice was collected.**
A further opportunity to sample a dust plume occurred on 16th October 2017, when particles of Saharan dust and
smoke from Iberian forest fires were transported across the UK. An instrumented radiosonde was prepared rapidly
and launched to allow the plume to be sampled in situ (Harrison et al, 2017c). Figure 15 shows a comparison data
from a surface ceilometer, and the radiosonde's charge and turbulence data. Turbulence was detected at the base and
top of the plume, with charge variability throughout the plume. The co-located charge and turbulence supports the
hypothesis of charge generation from in-plume turbulence, as for the Eyjafjallajökull plume.

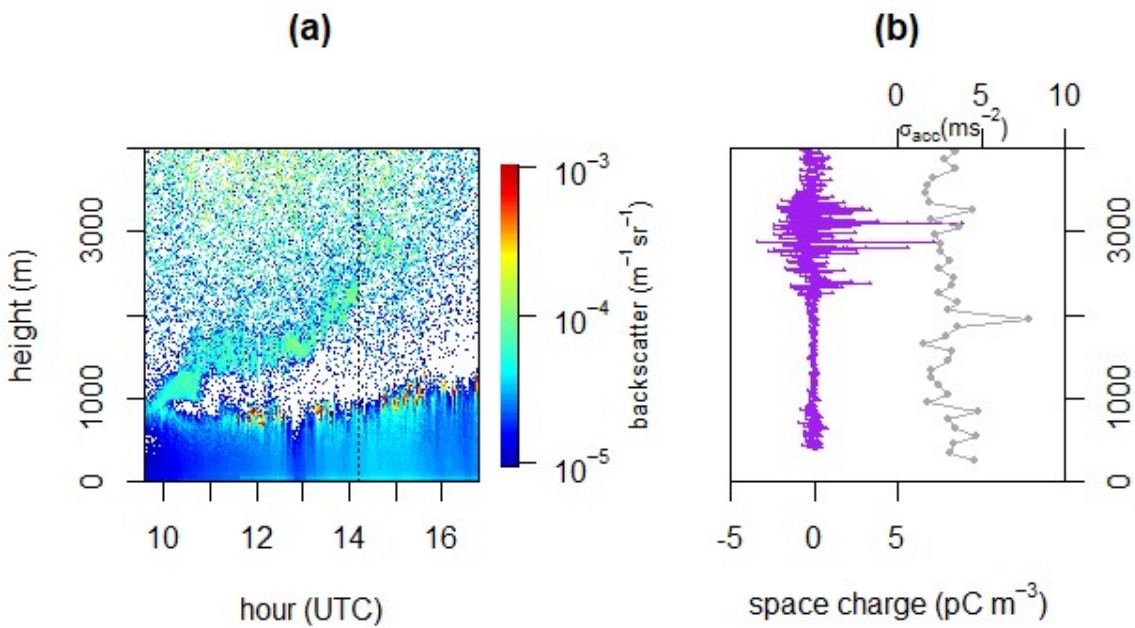

**(a)** **(b)**


**Figure 15 Dust cloud over Reading, 16th October 2017, sampled by an instrumented radiosonde released at 1412 UTC. (a)**
**Ceilometer backscatter profiles around the time of the sounding, with broken cloud around 1000 m through which some**
**dust fall occurred. (b) Charge profile (lines) determined by the balloon electrometer and the standard deviation of the**
**vertical acceleration (lines and dots) encountered by the balloon package, calculated in vertical steps of 100 m (adapted**
**from Harrison et al, 2017)**

**4.8 Coordinated use of research radiosondes**
One of the anticipated benefits in using standard radiosondes to carry enhanced instrumentation was to allow existing
sites to provide additional soundings, launched by those already familiar with radiosondes. An opportunity for this
occurred with the solar eclipse of March 20th, 2015. Eclipses are of course well predicted astronomically, but the
meteorological circumstances, and how much cloud may occur, is often less predictable. Radiosondes provide the
possibility of carrying small science packages above the cloud, and into a potentially more consistent measurement
environment.

The experiment envisaged was comparison of predicted and measured solar radiation during the eclipse, using the
radiosonde solar radiation sensor of Nicoll and Harrison (2012). For this, however, as eclipse opportunities are rare,
using multiple launch sites seemed prudent, in case one launch failed due to a balloon burst or instrument malfunction.
Hence, as well as from Reading, further coordinated launches on the eclipse path were arranged from the Met Office
at Lerwick and the Icelandic Met Office at Reykjavik. Ultimately, three solar radiation radiosondes successfully
provided measurements aloft during the eclipse (Harrison et al, 2016b). Figure 16a, b and c show the predicted solar
radiation changes during the eclipse at Reykjavik, Lerwick and Reading, and the consistency with the solar radiation
measured by the solar radiation radiosondes launched above these sites (Figure 16d, e, f) respectively. The partial
aspect of the eclipse at Reading is especially evident (Figure 16f).

These above-cloud eclipse measurements demonstrate the ease of deployment of the radiosonde systems at other sites.
Since then, many successful soundings carrying enhanced sensors have been carried out in Antarctica and in the United
Arab Emirates, with, as for the eclipse measurements, the generous support of colleagues working with radiosondes.

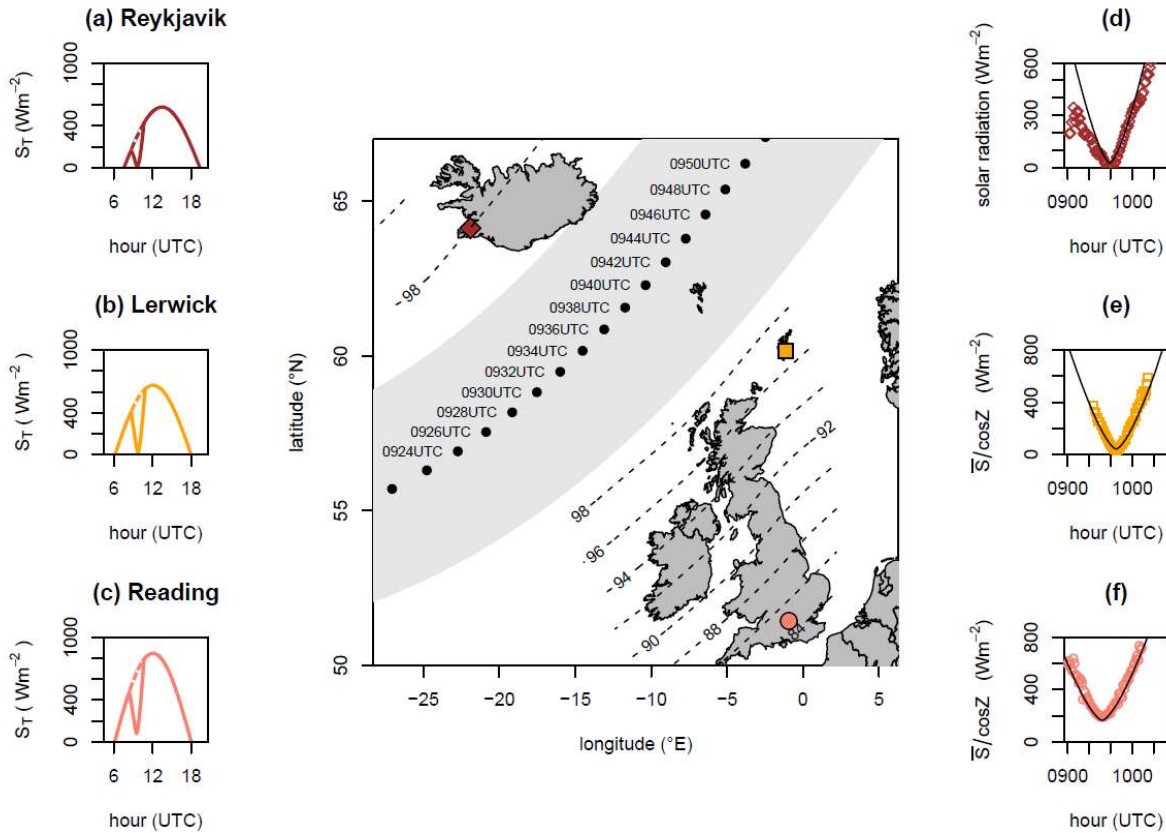


**Figure 16 Solar radiation variations during the 20th March 2015 eclipse, predicted for (a) Reykjavik, (b) Lerwick and (c)**
**Reading, and (d), (e), (f) measured by radiosonde above the same three sites respectively. The central panel shows the**
**region of totality (grey band) with timings and the partial eclipse fractions. Reykjavik (diamond), Lerwick (square) and**
**Reading (circle) are marked. (Adapted from Harrison et al, 2016b).**
**5.    Summary of layer cloud charge observations**
The original goal in the late 1990s of obtaining more information on the charge associated with extensive layer clouds,
which can now be revisited. With the range of sensors developed, and hundreds of instrument packages deployed,
positions of cloud boundaries can now be accurately determined, along with the background ionisation environment,
in-cloud turbulence and charge. The expectation from electrostatic theory, outlined in section 4, was that the upper
and lower boundaries of extensive layer clouds would carry positively and negatively charged respectively.

From soundings in Europe and Antarctica, charging on the upper and lower boundaries of extensive layer clouds has
been confirmed to be widespread (Nicoll and Harrison, 2016) and should be expected to be a global phenomenon. On
average, an upper positive charge and lower negative charge does emerge (Figure 17, upper panel). Several factors
influence this however, specifically the current flowing through the cloud, the meteorological conditions defining the
cloud edge properties and turbulent mixing within the cloud, and the background ionisation environment. Whilst
important conceptually, the idealised one-dimensional electrostatic case has been found to be incomplete, as it neglects
vertical charge exchange by mixing and variability in the vertical gradient from cloudy air to clear air.

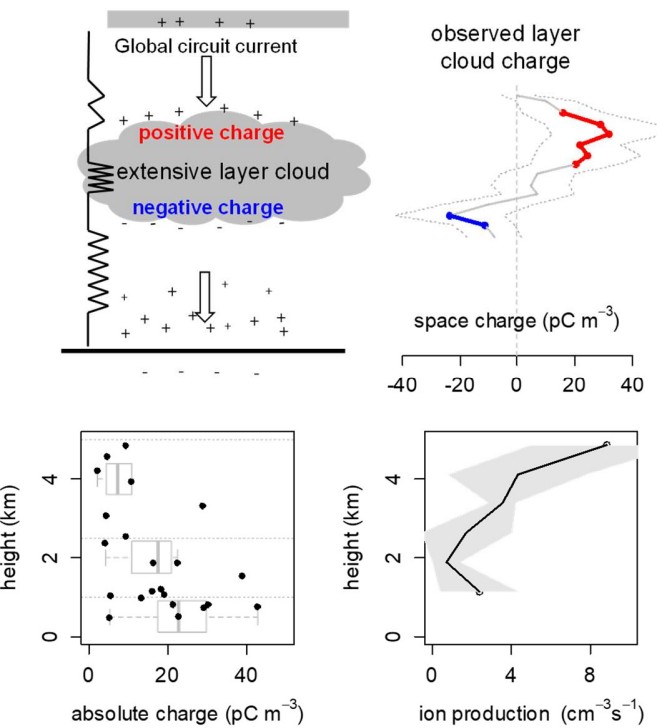


**Figure 17 Summary of radiosonde findings concerning extensive layer clouds.** *Upper panel:* **(left) Expected vertical charge structure around a layer cloud, arising from vertical current flow through a step change in air conductivity at the horizontal cloud boundaries, and (right) observed charge structure from soundings in both hemispheres.** *Lower panel:* **(left) absolute charge observed on layer clouds and (right) variation in ion production rate with height, averaged from the same sites as the cloud charge measurements. (Adapted from Nicoll and Harrison, 2016).**

A further finding is that less absolute charge is present at the horizontal cloud boundaries of higher (>2 km) clouds
than for lower (< 2 km) clouds, due to the greater air conductivity with height from increased cosmic ray ionisation
(Figure 17, lower panel). Low clouds often form where the vertical profile of air conductivity is at its minimum, and
hence where the rate of charge leakage away from the droplets is at its least. Cosmic ray ionisation, which is the
principal source of air conductivity above the surface, therefore does have a modifying influence on the charge at layer
cloud boundaries. The cloud boundary charge will, however, also respond to changes in the global circuit current
flowing through the cloud. Neutron Monitor data may not be a good predictor for this, as, quite apart from the
meteorological variability also influencing the cloud, Geigersondes have shown neutron monitor data to be poorly
correlated with the ionising environment at the typical altitudes of low clouds. However, there is no reason to doubt
that solar variability does modulate atmospheric electrical parameters in the troposphere, which are coupled directly
into the electrical properties of low cloud.

The full effects of droplet charge on cloud processes are still being evaluated. Charge is known to affect droplet
collisions, and droplet population modelling shows that droplet growth rates can be enhanced as a result (Harrison et
al, 2015; Ambaum et al 2021).

**6.    Conclusions**
Radiosondes are widely used by meteorological services worldwide, but are currently under-exploited as a platform
for research measurements beyond obtaining standard meteorological quantities. Additional measurements can readily
be obtained at low cost if standard radiosondes are suitably modified, ensuring that the core meteorological data is
unaffected. An effective way to do this has been through providing a standard interfacing sub-system (PANDORA in
the cases described), which can be adapted as commercial radiosondes evolve or are superseded, whilst retaining the
same connections to the existing individual sensor systems. Meteorological radiosondes can also provide a rapid
monitoring capability with many potential launch sites available. This could be in response to monitoring sudden dust
clouds or space weather events, or in emergency during volcanic ash or radioactivity dispersal, which also minimises
exposure to operators in a hazardous environment. Many sensors suitable for such work already exist, and if they do
not, application-specific devices can constructed, as demonstrated here. Alternatively, some quantities may be
obtained serendipitously, by repurposing sensors mass-produced for other applications, such as the accelerometers
used for turbulence detection.

Each atmospheric sounding represents a single measurement, but, unlike a laboratory measurement, it cannot be
repeated due to atmospheric variability. Continuing data reception after the ascent to capture descent data offers one
way in which a second sample can be obtained, usually in similar circumstances. Including multiple corroborating
sensors on each instrument package has also been found to be highly valuable, as the additional information provided
can help distinguish genuinely exceptional data from the merely anomalous. However, the constraints of limited mass
and size, finite power and restricted telemetry can force compromises in what can be combined with what. In some
respects, these design considerations mimic the tight engineering specification and need for success of a small space
mission. Similarly, rare or transient atmospheric circumstances, such as solar eclipses or a dust cloud, will be
unforgiving of system failures.

In summary, enhancing meteorological radiosondes as a research strategy has proved successful. It has extended
beyond the original expectations to include locating a volcanic ash plume in a national emergency, detecting solar
energetic particles entering the lower troposphere and, of special relevance here, offered new insights into data from
the Huygens descent probe in Titan's atmosphere.

Having begun with Christiaan Huygens' words, it seems fitting to close with a further quotation, in which, perhaps, there is a prescient hint of the value of research radiosondes:

"We may mount from this dull Earth; and viewing it from on high, consider whether Nature has laid out all her cost and finery upon this small speck of Dirt" (Huygens, 1722).

## Acknowledgements

I am grateful to many valued co-workers who have helped me considerably with developing instruments, performing experiments and discussing the results. I cannot list them all, but, for their determination with the ups and downs of experimental work, and especially radiosondes, I would like particularly to thank Keri Nicoll, Karen Aplin, Alec Bennett, Graeme Marlton and Martin Airey. Charles Clement (then at Harwell Laboratory) and Helen ApSimon (Imperial College) greatly encouraged my interest in atmospheric electricity. Some of the work described has been funded by STFC (ST/K001965/1), NERC (NE/H002081/1, NE/P003362/1, NE/L011514/1) and the Royal Society Paul Instrument Fund.

## Code/Data availability

The results presented have previously appeared in the publications referenced.

## Competing interests

The University of Reading is making the Geigersondes available commercially. There are no other competing interests.

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
