# Peer review of "Measuring electrical properties of the lower troposphere using enhanced meteorological radiosondes"

_Geoscientific Instrumentation, Methods and Data Systems, 2021_

## Referee Comment (RC2)

Review details:

Scientific significance: Excellent  (1)

Scientific quality:  Excellent (1)

Presentation quality: Excellent (1)

The presented paper ''Measuring electrical properties of the lower troposphere using enhanced meteorological radiosondes'' is a detailed follow-up on the related EGU medal lecture. It summarizes in a comprehensive way various possibilities of utilizing standard meteorological radiosonde balloon flights for additional science objectives. This approach can extend the balloons' main objectives of providing routine in-situ measurements of the atmospheric parameters pressure, temperature and humidity into areas interesting for basic research like understanding the charge flows in cloud layers, but also for emergency observations like ash transport after volcano eruptions. With the presented standardized low-cost approach a wide range of instrumentations could be prepared in advance and kept in stock at the institutes responsible for these balloon flights in case a fast emergency observation is needed or special atmospheric conditions might offer suitable observation possibilities. The paper offers sufficient detailed examples for the implementation of such instrumentation to allow the readers an evaluation of the possible suitability of such instruments for their own scientific or operational goals. It is therefore well suited for publishing in the GI journal.

While the paper is rather long and detailed and perhaps a bit difficult to digest, its clear structure and presentation of both historical background and modern technological implementation approaches for a wide range of different observation parameters make the reading interesting from start to end.

The graphic presentation of example measurements and comparative ground measurements of the same parameters show the suitability of the selected approach.

As the described instruments are very limited in available energy and data bandwidth and are supposed to work in an environment with potentially heavy mechanical disturbances and at low temperatures, an additional information about the worst case parameters would be interesting: lowest operational temperatures, maximum tested shock levels, power consumption per hour of operation and maximum duration with the used batteries.

Technical comments:

Line 151: , implementing bespoke analogue signal conditioning circuitry is still necessary.

I was wondering if you could find a different expression for the word ''bespoke''. (I as a none-English native speaker had to ask my wife, fortunately an English language teacher, what the meaning might be. Others might have similar problems.)

For many listed references the DOI is given. It would be helpful for the reader if all DOI references would be presented in the same complete web-link form, starting with https://doi.org/ While this is partly done, it is missing on lines 753, 757, 793, 795, 803, 813, 817, 825, 827, 829, 835, 881, 884 and 898. I suggest to add the full link annotation as a curtesy to the reader.

The link on line 921 is outdated. The referenced Vaisala document can now be found via

https://www.vaisala.com/sites/default/files/documents/WEA-MET-RS41-Datasheet-B211321EN.pdf

Please add also the standard comment for web page references, the date when it was last checked.

---

## Author Response (AR1)

**Response to referee comments on:** *Measuring electrical properties of the lower troposphere using enhanced meteorological radiosondes* **(gi-2021-26)**

I am most grateful to both reviewers for their careful reading of the manuscript and the thoughtful comments made. I have made changes in response to these, which are marked on the manuscript in red. I have also added some very minor clarifications and corrections of my own (lines 37-38, footnotes 10 and 11).

**Response to referee 1**

Thank you for the very positive comments and appreciation of the manuscript.

**Response to referee 2**

Thank you for the careful consideration of the manuscript. I greatly appreciate that it is concluded to be suitable for the journal.

*…an additional information about the worst case parameters would be interesting: lowest operational temperatures, maximum tested shock levels, power consumption per hour of operation and maximum duration with the used batteries.*

Sentences have been added at Line 311 to mention measured internal temperature, observed acceleration and battery life.

*Line 151: , implementing bespoke analogue signal conditioning circuitry is still necessary.*
*I was wondering if you could find a different expression for the word "bespoke". (I as a none-English native speaker had to ask my wife, fortunately an English language teacher, what the meaning might be. Others might have similar problems.)*
This word is no longer used. Where needed, it has been changed to "customised", and "for a chosen application", which I hope is more accessible*.*

*It would be helpful for the reader if all DOI references would be presented in the same complete web-link form, starting with https://doi.org/ While this is partly done, it is missing on lines 753, 757, 793, 795, 803, 813, 817, 825, 827, 829, 835, 881, 884 and 898.*

These have been corrected, and many other links are now given where possible.

*The link on line 921 is outdated. The referenced Vaisala document can now be found via* [https://www.vaisala.com/sites/default/files/documents/WEA-MET-RS41-Datasheet-B211321EN.pdf](https://www.vaisala.com/sites/default/files/documents/WEA-MET-RS41-Datasheet-B211321EN.pdf)

Thank you - this has been changed.